# A Scoping Review of the Global Distribution of Causes and Syndromes Associated with Mid- to Late-Term Pregnancy Loss in Horses between 1960 and 2020

**DOI:** 10.3390/vetsci9040186

**Published:** 2022-04-13

**Authors:** Claudia M. Macleay, Joan Carrick, Patrick Shearer, Angela Begg, Melinda Stewart, Jane Heller, Catherine Chicken, Victoria J. Brookes

**Affiliations:** 1School of Agricultural, Environmental and Veterinary Sciences, Faculty of Science and Health, Charles Sturt University, Wagga Wagga, NSW 2678, Australia; pshearer@csu.edu.au (P.S.); jheller@csu.edu.au (J.H.); 2Equine Specialist Consulting, Scone, NSW 2337, Australia; joancarrick@bigpond.com; 3Diagnostic Laboratories Pty Ltd., New Lambton, NSW 2305, Australia; apbegg@bigpond.com; 4Starling Scientific, Pearl Beach, NSW 2256, Australia; starling.scientific18@gmail.com; 5Graham Centre for Agricultural Innovation, NSW Department of Primary Industries, Charles Sturt University, Wagga Wagga, NSW 2678, Australia; victoria.brookes@sydney.edu.au; 6Scone Equine Hospital, Scone, NSW 2337, Australia; catherine.chicken@sconeequine.com.au; 7Sydney School of Veterinary Science, Faculty of Science, The University of Sydney, Camperdown, NSW 2008, Australia

**Keywords:** equine reproduction, equine pregnancy loss, abortion, veterinary pathology, veterinary epidemiology, scoping review

## Abstract

Equine pregnancy loss is frustrating and costly for horse breeders. The reproductive efficiency of mares has significant implications for a breeding operation’s economic success, and widespread losses can have a trickle-down effect on those communities that rely on equine breeding operations. Understanding the causes and risks of equine pregnancy loss is essential for developing prevention and management strategies to reduce the occurrence and impact on the horse breeding industry. This PRISMA-guided scoping review identified 514 records on equine pregnancy loss and described the global spatiotemporal distribution of reported causes and syndromes. The multiple correspondence analysis identified seven clusters that grouped causes, syndromes, locations and pathology. Reasons for clustering should be the focus of future research as they might indicate undescribed risk factors associated with equine pregnancy loss. People engaged in the equine breeding industry work closely with horses and encounter equine bodily fluids, placental membranes, aborted foetuses, and stillborn foals. This close contact increases the risk of zoonotic disease transmission. Based on this review, research is required on equine abortion caused by zoonotic bacteria, including *Chlamydia psittaci*, *Coxiella burnetii* and *Leptospira* spp., because of the severe illness that can occur in people who become infected.

## 1. Introduction

Equine industries exist in most countries worldwide, with horses used for various purposes, including working animals, meat production, or facilitating entertainment and recreation. The foundations of these equine industries are those that breed horses. The sole purpose of breeding the mare is to produce a healthy foal. Equine pregnancy loss is a serious issue to horse breeders because each broodmare represents a substantial long-term investment of time and money. A study on mare profitability from Kentucky found that a mare must produce a live foal six out of seven years to yield a positive financial return from her initial purchase, with profitability the highest among mares with the highest monetary value [1]. Pregnancy loss reduces equine reproductive efficiency and is commonly measured using live foal, pregnancy and pregnancy loss rates [2]. In countries with large Thoroughbred industries, foaling rates have increased over the past 50 years. Data from the Australian Studbook reveals that annual foaling rates have almost doubled from 38.2% in 1979 to 66.69% in 2018. Likewise, in Kentucky, USA, foaling rates have increased from 66.1% in 1988 to 78.3% in 2004 [3,4], and in Newmarket, UK from 70% in 1972 to 82.7% in 1999 [5,6]. These increases can be attributed to better veterinary intervention and treatments, such as the reduction in twin pregnancies due to ultrasound scanning during early pregnancy and the manual crushing of one conceptus [7]. Reported rates of loss show wide geographic variation. Reported annual estimates of the proportion of lost pregnancies in Thoroughbred mares ranges from 7.9% in the UK [8], to 15.2% in Ireland [9] and 12.9% in Kentucky, USA [3]. Causes of equine pregnancy loss include twinning, foetal and placental abnormalities, maternal disease, and viral, bacterial and fungal infections [10,11,12,13], but the global variation of reported causes is not well documented. Therefore, the objective of this scoping review was to document the reports about causes and syndromes associated with equine pregnancy loss globally. Scoping reviews provide a broad overview of current evidence, describe the body of literature about a topic and highlight gaps in research activity, especially for topics with a large and diverse evidence base [14]. Specifically, this review aimed to identify the currently available, published literature on equine pregnancy loss and extract data to categorise the spatiotemporal distribution of reported causes and syndromes globally. Describing the global variation in reported causes of equine pregnancy loss is an important first step for investigating regional determinants because it provides a baseline of current knowledge and highlights differences in diseases, causes and syndromes being reported. Therefore, this review provides a foundation of knowledge of equine pregnancy loss globally and identifies bodies of evidence suitable for future systematic reviews.

## 2. Materials and Methods

### 2.1. Protocol

This scoping review was conducted according to the Preferred Reporting Items for Systematic reviews and Meta-Analyses extension for Scoping Reviews, PRISMA-ScR, guidelines [15]. The review protocol included three levels: Level 1, screening on the title and abstract; Level 2, screening on full record; Level 3, data charting. All levels were conducted in the web-based review platform sysrev (www.sysrev.com, accessed on 29 January 2020). The scoping review protocol is included in the Supporting Information. The term ‘record’ was used throughout this paper to describe any bibliographic citation captured in the searches.

### 2.2. Eligibility Criteria

Eligible records were any which contained information on the epidemiology and pathology associated with mid to late-term pregnancy loss in horses, were primary sources (whether peer-reviewed or not), published in English after 1960, and for which the full text was available. The records were sourced from 1960 due to changes in histopathology identification and some laboratory methods. The population of interest in the study were horses and ponies (*Equus caballus*). Free-living equines were feral, wild or unowned, independent of intentional feeding or husbandry from humans. Domestic equines were defined as animals bred for domestic purposes or those wholly or partially dependent on human husbandry.

### 2.3. Definitions of Equine Pregnancy Loss

Pregnancy loss was categorised into four stages based on the development of the foetus. Embryonic loss was categorised from 1–45 days, though the equine embryo transitions from the yolk sac to the allantoic sac by day 40 [16]. Current industry practice is to perform ultrasound scanning between days 40–45 to confirm that the pregnancy had reached past the embryonic phase. Early foetal loss was measured from the post scanning period of day 46–150 days when foetal development is predominantly characterised by the maternal maintenance of pregnancy from endometrial cups [17]. Mid-gestational foetal loss were identified as those from 151–270 days, which is a period of growth, both in foetal size and weight. Late-term foetal loss were those >270 days. The final stage of equine foetal development ensures that the equine foetus is prepared for an independent life outside of the uterus and is defined by the maturation of the hypothalamic–pituitary–adrenal axis (HPA), which occurs within the last 15–20 days before birth [18,19]. Stillbirths were classified as late-term foetal losses. Dystocia, which is foetal death from difficulty during birth, and neonatal death of foals were excluded.

### 2.4. Search Strategy

An initial search of six electronic databases; CAB Abstracts, MEDLINE, ProQuest, PubMed, Scopus, and Web of Science, was conducted in April 2020. The initial search was performed using the following combination of terms:


*Population: equine* OR horse* OR mare* OR thoroughbred* OR broodmare* OR pony OR ponies*

*Topic: “pregnancy loss” OR abortion OR “fetal loss”OR “foetal loss” OR “fetal mortality” OR “foetal mortality” OR “reproductive loss” OR “reproductive waste” OR “reproductive wastage” OR “premature labor” OR “premature labour” OR “premature delivery” OR placenta OR “reproductive pathology” OR stillbirth OR “perinatal mortality”*


As a complementary search, the first 100 results from the search engine “Google Scholar” were screened to identify any relevant records that were not previously captured. The search of databases for relevant records published since April 2020 was repeated on 6 October 2020, these records were included in the finial 514 records used in the scoping review. Other records potentially relevant to our study were included in the review process using the same methods as records identified in the initial search. All records were exported into the citation manager software Endnote. Duplicate records were identified in Endnote and removed before being uploaded to sysrev.com (www.sysrev.com, accessed on 29 January 2020).

### 2.5. Selection of Relevant Records—Levels 1 and 2

Before screening commenced at Levels 1 and 2, agreement tests were performed in which the authors all reviewed the same randomly selected records (Level 1 = 50 records, and Level 2 = 30 records). Conflicts of opinion about inclusion and exclusion of records were discussed to achieve consensus between reviewers. After these discussions, the forms were refined to improve clarity and consensus before the full review for each level. Two reviewers assessed each record at Levels 1 and 2. The first screening level was undertaken to eliminate irrelevant articles based on the abstract (or the title when the abstract was not available). If there was a conflict of opinion about including or excluding a record, the two reviewers discussed whether to include it in the next level. If either reviewer was uncertain, the record was included in Level 2. During Level 2, records were excluded if the full text could not be sourced, if the text was not in English, or it did not contain any information consistent with the study objectives. The remaining records progressed to Level 3. At Level 2, conflicts of opinion and record eligibility were assessed by a third reviewer (VB, JC or AB). The record progressed to Level 3 if two of the three reviewers considered the record potentially relevant.

### 2.6. Verification of Search Strategy

Our search strategy was verified by screening the bibliography list of 5 key reviews published between 1960–2020. After removing duplicates, a total of 115 records remained for verification. Forty-two were excluded based on title and abstract, one because the record was not in English and 16 because the record was published before 1960. The remaining 56 records had all been captured in Level 1. Additionally, a specialist in equine medicine (JC) reviewed the list for missing relevant references.

### 2.7. Data Charting and Extraction—Level 3

All records that met the eligibility criteria from Level 2 advanced to Level 3 for data charting and extraction. The form for Level 3 was used as a data charting tool to standardise the extracted data (see Appendix A). Data (topics of interest) extracted were the year of publication, the location and year of the study, the population of equines, causes and syndromes of equine pregnancy loss, and whether the gross pathology associated with equine pregnancy loss and histopathology associated with equine pregnancy loss were reported. For all questions, a ‘not specified’ option was available. Due to the Level 3 form being longer and more complex than the previous level, the agreement was assessed on four records, selected randomly from records included in Level 3. Conflicts were discussed, and the form was modified to improve clarity and increase agreement, after which Level 3 reviewing was conducted.

### 2.8. Data Synthesis to Identify Cluster between Topcis of Interest

Data from Level 3 were downloaded into Microsoft Excel (version 2013). Descriptive statistics from Level 3 were analysed using the statistical program R [20] and the packages plyr [21], tidyverse [22] and ggplot2 [23]. A multiple correspondence analysis (MCA) was performed on the records captured in Level 3 to describe, visualise and identify potential clusters of topics of interest. Records that had documented pregnancy loss from multiple causes, different locations and at various stages of gestation were reviewed and divided so that causes, location, and gestation were accurate. The active variables in the MCA were the reported causes of equine pregnancy loss. The supplementary quantitative variables in the MCA were location, breed, stage of gestation and pathology investigation. The supplementary qualitative variable in the MCA was the decade of publication. The MCA was conducted in R [20] using the packages FactoMineR [24] and factoextra [25]. Multiple cor-respondence analysis was conducted as a pre-processing step before hierarchical cluster-ing was performed using the HCPC function (Hierarchical Clustering on Principal Com-ponents) from the FactoMineR package [24,25]. The number of clusters selected for the cluster analysis were based on the bar plot of the gain of within-inertia produced by the hierarchical tree dendrogram [26].

## 3. Results

### 3.1. Level 1 Screening

After removing duplicates from all searches, including the verification strategy, 8854 records were identified (Figure 1). Most records in Level 1 were excluded because they did not include data on equine pregnancy loss (69%, *n* = 6032), were published before 1960 (10%, *n* = 893), or were not primary information (4.5%, *n* = 396) (Figure 1). A specialist in equine medicine (JC) recommended additional records (*n* = 111) not captured in the Level 1 search; these were industry and workshop publications not listed on journal databases or Google Scholar.

### 3.2. Level 2 Screening

In total, 1533 records were included for the full-text screening during Level 2 (Figure 1). The most common reason for exclusion was that the record did not contain relevant information (32%, *n* = 427). Records were excluded if they were not in English (21%, *n* = 325). Of those records not in English, there were 29 different languages recorded. The most common languages other than English were German (23%), Polish (13%) and French (12%). Records were also excluded because the full text could not be procured (12%, *n* = 177), the record was not primary information (*n* = 67), they were published before 1960 (*n* = 9) or were duplicates (*n* = 14). No new records were identified in the search verification strategy. The repeated search for recent publications returned eight new records which were included in Level 3. Five hundred and fourteen records were eligible for data extraction by charting and data synthesis in Level 3.

### 3.3. Data Charting

A total of 514 records were included in Level 3 (Figure 1). A spreadsheet of the Level 3 studies and the data charted for each study is included in the Appendix A. Records sourced from peer-reviewed journals (*n* = 384) were predominantly from the Equine Veterinary Journal (*n* = 35), the Veterinary Record (*n* = 24), the Journal of Veterinary Diagnostic Investigation (*n* = 24) and the Journal of Equine Veterinary Science (*n* = 23). Other records included grey literature (*n* = 95), conference proceedings (*n* = 27), non-journal reports (*n* = 2), thesis chapters (*n* = 2), science forum websites (*n* = 2), government reports (*n* = 2). Records were published from 1960–2020, with data collected since 1930. Records collected data over a period of 1–65 years (mean 4.5 years) (Figure 2).

Most records were descriptive studies (72%, *n* = 370), followed by analytic observational studies (11%, *n* = 58), outbreak investigations (6%, *n* = 31) and comparative experimental studies (5%, *n* = 29). Records in the category of ‘other’ (6%, *n* = 30) included studies on the diagnostic investigations of specific pathogens able to cause equine abortion or pathogens discovered in abortion cases. The country of data collection was recorded in 446 records (87%) (Figure 3). Ninety-four records had data collected from more than one country (18%). Of these, 225 records (50%) included locations within countries. Most records collected data from Europe (*n* = 409), North America (*n* = 203) and Asia (*n* = 102).

The main reported cause of pregnancy loss was EHV-1 which was documented in 223 records (43%). Other common causes that were reported were placentitis (17.7%, *n* = 91), leptospirosis (14%, *n* = 72), twinning (9.7%, *n* = 50), congenital abnormalities (7.2%, *n* = 37), EHV-4 (6.2%, *n* = 32), umbilical cord torsion (6.2%, *n* = 32) and equine amnionitis and foetal loss/mare reproductive loss syndrome (EAFL/MRLS) (5.4%, *n* = 28). Ninety-six publications recorded no specific cause of pregnancy loss and were classified as ‘unspecified cause’ (18%). Miscellaneous pregnancy loss causes accounted for 29% of reported causes (*n* = 149). The most commonly reported cause in this category was abortion due to infections from Streptococcus equi subsp. zooepidemicus (*n* = 28), Escherichia coli (*n* = 18), and equine viral arteritis (*n* = 20). There were 18 records of mares losing a pregnancy due to hydroallantois or hydrops. Premature placental separation was reported as a cause of pregnancy loss in 14 records. Other miscellaneous causes were rare diseases (for example, Bunyamwera virus), uncommon genetic abnormalities (for example, Lymphosarcoma) and unusual bacterial infections (for example, Hafnia alvei).

When examining records for temporal trends in reported causes over time (Figure 4), the earliest reported causes in the 1930s were twinning, umbilical cord torsion and unspecified causes. EHV-1 appears in records in the 1940s, congenital abnormality appears in the publications in the 1950s, and the first descriptions of placentitis and leptospirosis appear in the 1960s. Ascending placentitis and nocardioform placentitis both entered into the literature in the 1990s. Records on twinning, leptospirosis, EHV-4 and congenital abnormalities peaked in the 1990s. Chlamydia and EAFL/MRLS entered the literature in the 2000s, with records on EHV-1, unspecified causes and other miscellaneous causes peaking in the 2000s.

The stage of gestation within which the abortion or stillbirth occurred was recorded reported in 301 records (59%). Early-, mid- and late-gestation loss was reported in 66 (22%), 181 (60%) and 206 (68%) records, respectively. Although not an inclusion criterion, embryonic pregnancy loss was reported in 25 records (8%) in conjunction with losses at other stages of pregnancy. During early and mid-gestation, the most frequently reported cause of pregnancy loss were miscellaneous causes (*n* = 19 and *n* = 74, respectively). The next most frequent cause reported during mid foetal gestation was EHV-1, followed by placentitis and leptospirosis (*n* = 26 and *n* = 21, respectively). In late gestation, EHV-1 was the most frequently reported cause (*n* = 64), followed by miscellaneous causes (*n* = 59), placentitis (*n* = 39) and leptospirosis (*n* = 26) (Figure 5).

Seventy-five different breeds of horses were reported in the scoping review. Two hundred and eleven records contained information on Thoroughbreds (41%), with 128 records collecting data only on Thoroughbreds. Other breeds that featured prominently in records included Standardbreds (*n* = 49), Quarter horses (*n* = 44) and Arabian horses (*n* = 35). One hundred and seventy-four records (34%) provided no information on the horse breed. The gross pathology of the foetus was recorded in 30% of publications (*n* = 156), while the gross pathology of the foetal membranes was recorded in 24% of publications (*n* = 125). The histopathology of the foetus was recorded in 28% of publications (*n* = 144), while the histopathology of the foetal membranes was only recorded in 20% of publications (*n* = 105).

### 3.4. Clustering of Topics of Interest

#### 3.4.1. Assessment of Outliers

Preliminary analysis of the MCA factor maps (Appendix A) indicated 50 records that were dispersed (without visible clustering). When these records were reviewed, they were predominantly longitudinal studies and case studies of rare causes of equine pregnancy loss (for example, hydrocephalus and Warmblood fragile foal syndrome) and were removed from further analysis. The factor maps from the first six dimensions of the MCA analysis, the eigenvalues and the hierarchical tree dendrogram are included in the Appendix A.

#### 3.4.2. Multiple Correspondence Analysis

The first six MCA dimensions explained 45.23% of the variability among records and were retained for the analysis (Appendix A). The variables with the strongest contribution to Dimension 1 were ‘no EHV-1 recorded’, ‘unspecified cause’, twinning, ‘other bacterial cause recorded’, placentitis, and leptospirosis. The strongest variables contributing to Dimension 2 were EHV-1, ‘the cause of pregnancy loss was diagnosed’, ‘no twinning’ and ‘no other bacterial cause was recorded’ (Appendix A). The variables that had the strongest contribution to Dimension 3 were fungi, protozoa, umbilical cord torsion, Chlamydia and EHV-1. The supplementary variables with a stronger association to Dimension 3 were gross pathology on the foetus and foetal membranes. The variables with the strongest contribution to Dimension 4 were pregnancy loss due to ‘other placental causes’ and ‘other environmental causes’. The supplementary variables associated with Dimension 4 were ‘gross and histopathology on the foetal membranes’ (Appendix A). Dimension 5 variables with the strongest contribution were leptospirosis and protozoa. Variables with the strongest contribution to Dimension 6 were ‘no leptospirosis’, EAFL/MRLS, fungi and ‘other foetal conditions’. The supplementary variables with the strongest association to Dimension 6 were the global region recorded as Australia and ‘histopathology on the foetal membranes’ (Appendix A).

#### 3.4.3. Hierarchical Clustering

Seven clusters (Figure 6) were identified based on the bar plot of the gain of within-inertia produced by the hierarchical tree dendrogram (Appendix A). Contributing factors are summarised in Table 1. Cluster 1 (*n* = 553) contained records reporting EHV-1 from Europe. Breed of horse and stage of gestation were not documented, and neither were gross pathology or histopathology of the foetal membranes. Cluster 2 (*n* = 120) contained records on leptospirosis in Thoroughbred horses located in North America. Foetal histopathology had been conducted. Cluster 3 (*n* = 194) contained records on the cause of pregnancy loss from other viruses (not EHV-1) and other placental, maternal, and environmental causes. Pregnancy loss in these records occurred between mid and late gestation in non-Thoroughbred horses. Histopathology and gross pathology on the foetal membranes had been documented. Cluster 4 (*n* = 131) was constructed of records on abortion from caterpillars, including Equine Amnionitis and Foetal Loss and Mare Reproductive loss syndrome (EAFL/MRLS), with placentitis, other bacteria and other foetal causes, from North America. Records documented gross and histopathology on the foetal membranes and gross and histopathology on the foetus. Cluster 5 (*n* = 31) contained records of twinning and cases with no specific diagnoses in Thoroughbred horses in Asia. Pregnancy loss occurred in embryonic and early gestation and did not report EHV-1, and no histopathology conducted on the foetus. Cluster 6 (*n* = 37) included records on umbilical cord torsion, protozoa and placentitis in non-Thoroughbred horses in early gestation. The location of these cases was either not recorded or in South America. No EHV-1 was documented. Gross pathology on foetal membranes and histopathology on the foetus was included in these records. Cluster 7 (*n* = 20) consisted of records that included fungi, congenital abnormalities, chlamydia and placentitis with no EHV-1. Cases occurred during mid-gestation in non-Thoroughbred horses. In these records, gross pathology was conducted on the foetus and foetal membranes, and histopathology was conducted on the foetal membranes.

## 4. Discussion

This review provides a broad overview of the current literature on the causes and syndromes reported globally in mid to late-term pregnancy loss in horses, including the spatiotemporal distribution. An increase in reporting of equine pregnancy loss since 1960 was identified in the literature. The majority of data collected in the records from Level 3 came from countries with large Thoroughbred racing industries, which consequently have large Thoroughbred breeding populations, including the United States of America, the United Kingdom, Japan, France, Australia and South Africa [27]. The large volume of reporting of equine pregnancy loss is likely due to the value of Thoroughbred horse racing in these countries and the severe economic impact that reproductive disease outbreaks can have [28,29].

The causes of equine pregnancy loss documented in this review can be broadly divided into two categories: infectious and non-infectious causes. Infectious disease accounted for a large proportion of the records, demonstrating that these are the main causes of concern for equine pregnancy loss. Equine herpesvirus-1 (EHV-1) was the predominant reported cause of equine pregnancy loss throughout the study period for all stages of pregnancy, with most reporting from Europe as was identified from the MCA. Equine herpesvirus-1 is a highly infectious virus in horses, and abortion is a serious consequence of infection in the mare [30]. Horses that are infected become life-long carriers due to the ability for EHV-1 to remain latent, meaning horses with viral latency serve as a viral reservoir and can infect susceptible populations [31]. In unvaccinated herds with inadequate biosecurity, up to 75–80% of mares may abort from EHV-1 infection [32,33]. In major horse breeding areas, including the United States and the United Kingdom, cases of EHV-1 have decreased over time, which was reflected in the literature captured in this review (Figure 4). In Kentucky, USA, the annual rate of abortion from EHV-1 between 2016–2019 ranged from 1–5% [34], while the annual rate of EHV-1 abortion in Newmarket, UK between 2013–2017 ranged from 0.1–0.9% [34] with EHV-1 reported in 6.5% foetuses submitted for investigation in Newmarket between 1988–1997 [11]. These low figures can be attributed to a combination of vaccination and improved disease risk reduction, such as implementing farm quarantine and separating vulnerable animals from those that could be infectious [35]. However, a lapse in vaccination or farm biosecurity can result in abortion outbreaks, which have been reported recently in the UK and Poland [30,36]. Records on EHV-1 often had no gross pathology or histopathology documented but this was likely due to many of these records outbreaks being sourced from the Equine Disease Quarterly reports, which do not include these details. Overall, it appears that EHV-1 continues to be a major risk for equine pregnancy loss worldwide and that continued vaccination and biosecurity programs are required.

Placentitis is often described as the most common cause of infectious abortion [37]. However, it only accounted for 16% of reported pregnancy loss causes in this review. There may be underreporting on placentitis cases because it is frequently diagnosed following gross placental examination, and further cause-specific investigations requiring laboratory tests and histology are declined. Supporting this, placentitis reports were generally clustered with histopathology reports, suggesting that it is only reported in the literature in the context of a complete post-mortem examination. Pregnancy loss due to placentitis occurred in mid and late gestation, but a quarter of the records did not provide any information about the stage of gestation. Consistent with the underlying causes of placentitis, analysis of the clustering demonstrated that records of placentitis were grouped with bacterial, fungal, or protozoan infections. These findings, especially the lack of reports of placentitis without histopathology reports, suggest that surveillance systems of equine abortion in which putative causes are recorded, even in the absence of laboratory investigations, are necessary to provide accurate incidence of equine abortion. It is likely that the cost of full investigations is often prohibitive and that owners might only focus on ruling out communicable infectious causes such as EHV1.

Leptospirosis is a zoonotic bacterial disease with global distribution caused by the pathogenic serovars within the genus *Leptospira*. It was documented as the second most common cause of equine abortion in North America after EHV-1 and was clustered with Thoroughbreds and foetal histopathology. Only four percent of records documenting cases of leptospirosis were sourced from Europe. The reason for the clustering in reporting could be due to a combination of factors, including different serovars of *Leptospira* spp. in the US, climatic conditions and host animal species and their contact with horse populations [38]. However, this should not be a reason for complacency in countries that have not reported leptospirosis as a cause of equine abortions because the first documented case in the UK was recorded relatively recently, in 2009. In Australia, serological surveys have detected *Leptospira* spp. in healthy horses since 1982, but it has not yet been detected in aborted material [38,39,40,41,42].

An area identified as a gap in the literature from this review was a lack of reporting on equine abortion associated with the zoonotic bacterias *Chlamydia psittaci* and *Coxiella burnetii* [43,44]. Both can cause serious illness in people and both can infect people following exposure to infected placental membranes and aborted material [43,45]. However, only eight records on abortion associated with *Chlamydia psittaci* and two records on abortion associated with *Coxiella burnetii* were captured in Level 3. Abortion from *C. psittaci* was reported in Australia and Europe (France, Germany, Hungary and Switzerland) [38,46,47,48] and abortion associated with *C. burnetii* was reported in Australia and France [38,48], despite both bacteria being globally distributed. Disease due to *C. burnetii* is known as Q fever in people, and in Australia, it is the most commonly reported zoonotic disease. Exposure can occur from contact with infected animals or the inhalation of dust containing the bacteria [44]. *Coxiella burnetii* has been detected in the aborted foetal membranes, placental fluids, and vaginal mucus of cattle, sheep, and cats. Antibodies to the bacteria have been found in horse populations [49]. Based on the epidemiology, pathology and impact on human health, *C. burnetii* should be a pathogen considered in equine abortion investigation and is an area that requires further research and disease surveillance to understand incidence, determinants and risks to both horses and people.

In addition to *Chlamydia psittaci* and *Coxiella burnetiid*, a recent systemic review identified an additional 54 equine pathogens that had reported zoonotic transmission [50]. Of the 26 bacteria identified in the systemic review, there were 16 bacterial species also documented from this study as causes of equine abortion. These included *Actinobacillus* spp., *Brucella* spp., *Bartonella* spp., *Campylobacter* spp., *Escherichia coli*, *Ehrlichia*, *Enterococcus* spp., *Klebsiella* spp., *Listeria monocytogenes*, *Mycobacterium* spp., *Rhodococcus equi*, *Rickettsia* spp., *Streptococcus equi subsp. zooepidemicus*, *Salmonella* spp., and *Staphylococcus* spp. In addition, the protozoan *Toxoplasma gondii* and West Nile virus were also identified as zoonotic diseases with horse-human transmission and also recorded as causes of equine pregnancy loss. While zoonotic infections in people do not generally arise from exposure to aborted materials, it highlights the potential risk of zoonotic infection in people working closely with foaling mares, as these people are frequently in contact with bodily fluids, placental membranes and aborted foetuses, as well as stillborn foals and compromised neonatal foals. A recent study on personal biosecurity adopted by people working in the Australian thoroughbred breeding industry found that within the industry there needed to be greater awareness of zoonotic disease risks, the use of personal protective equipment and the need for a cultural shift within the industry to normalise of personal biosecurity [51].

The two main non-infectious causes of equine pregnancy loss identified in this review were twinning and umbilical cord torsion. Twinning clustered most strongly with Thoroughbred horses, and no histopathology was conducted. This reflected the types of studies involving Thoroughbred horses, which were often retrospective descriptive studies that did not record gross and histopathology findings. If left untreated, twinning is a significant cause of reduced reproductive efficiency in the mare and usually results in mid to late term abortion or stillbirth [52] because gestation proceeds normally until the twin foetuses begin to compete for placental and uterine space. Since twin foetuses have a reduced chorionic surface area, they suffer from growth retardation and inadequate nutrition due to placental insufficiency leading to abortion, premature birth and stillbirth [53]. Twinning had the largest decline as a reported cause of loss over the publication period. Twinning as a cause of loss accounted for 35% of reported losses in the 1960s but decreased to 3% in the 2000s. While Platt [54] reported that 29% of abortions in Thoroughbred mares in England between 1960–1967 were due to twinning, a similar investigation between 2013–2017 recorded no cases of twin abortions [34]. Other surveys on equine abortion have reported similar reductions over time [11,12,13,55,56,57]. This reflects the introduction of ultrasound examination of early pregnancy (<21 days), enabling the reduction in twin pregnancies in the mare [58,59] and demonstrates the benefit of veterinary care of mares during early pregnancy.

Umbilical cord torsion as a cause of pregnancy loss in domestic livestock is almost unique to the equine. In the early stages of gestation, the equine foetus can move within both the amniotic and allantoic sacs; only when the foetus increases in size towards late pregnancy is its mobility restricted [53]. Movement of the foetus can twist the umbilical cord; the limiting factor for foetal death is the extent of circulatory compromise or urinary retention that occurs [60]. In the current review, umbilical cord torsion was reported mainly from Europe and North America, occurring in both mid and late gestation. The length of the umbilical cords has been cited as a risk factor for umbilical cord torsion [61,62]. It is of interest that umbilical cord torsion records were clustered with records on protozoa and fungi, this might be due to similarities in the spatiotemporal clustering of these cases, although this was not apparent and could also be due to shared risk factors. Further investigation of the potential risk factors of equine pregnancy loss due to umbilical cord torsion would be worthwhile given its apparent frequency as a cause of pregnancy loss. Other non-infectious causes of pregnancy loss were reported sporadically throughout the study period, such as congenital abnormalities or maternal diseases. These included causes such as genetic diseases such as Warmblood fragile foal syndrome, and maternal diseases such as cancers and maternal reproductive problems such as uterine body pregnancies.

A limitation of this review was that only English records were captured, creating a lack of information on equine pregnancy loss in Africa, China, Eastern Europe, and Russia. Recent studies have estimated that China and Russia have a horse population of 3.47 million and 1.3 million, respectively. Most horses are used as work animals in rural areas or for horse meat production for human consumption [63,64,65]. Equine pregnancy loss outbreaks in these regions would cause an economic impact equal to that seen in Thoroughbred racehorses and should be considered an area of future research and disease surveillance. A limitation of this review this that reported cases may not necessarily represent incidence; instead, they map where reporting has occurred and what has been found using reproducible, rigorous review methods. Many cases of pregnancy loss are unlikely to have been investigated and therefore, do not contribute to reported cases. A challenge of this review was the balance between the breadth and depth of analysis, and consequently it is probable that despite a comprehensive search strategy some records may not have been captured in this review.

A valuable element of scoping reviews is identifying gaps in the current research. Gross pathology and histopathology were underreported in records, with less than a third of all records reporting it in the paper. There are two reasons for this finding, either the pathology and histopathology were never completed or not reported. Regardless of the reason, the outcome was a lack of reporting. The benefit of including the pathology and histopathology findings is that they support the clinical documentation of the cause of loss while adding to the knowledge and literature on equine abortion.

## 5. Conclusions

This scoping review identified the global spatiotemporal distribution of reported causes and syndromes of mid to late-term equine pregnancy loss. Seven global clusters of equine pregnancy loss were identified based on records, grouped by causes, syndromes, location, breed, and whether gross or histopathology was conducted and reported as part of the investigation. More detailed investigations of these clusters may reveal unknown risk factors associated with equine pregnancy loss. This review also identified two gaps in the current literature. Firstly, there is limited reporting of some zoonotic causes. People working in the horse breeding industry come into close contact with equine bodily fluids and infectious materials, placing them at an increased risk of infection with zoonoses. Identifying whether these zoonoses are underdiagnosed or emerging in equine populations, as well as their risk factors, should be a priority for researchers because of the potential for severe level of illness in people. Secondly, the lack of reporting of gross pathology and histopathology limits accurate clinical documentation of disease presentation and diagnosis. The findings of this review can be used to inform development of surveillance systems for equine abortion and important potential gaps in reporting.

## Figures and Tables

**Figure 1 vetsci-09-00186-f001:**
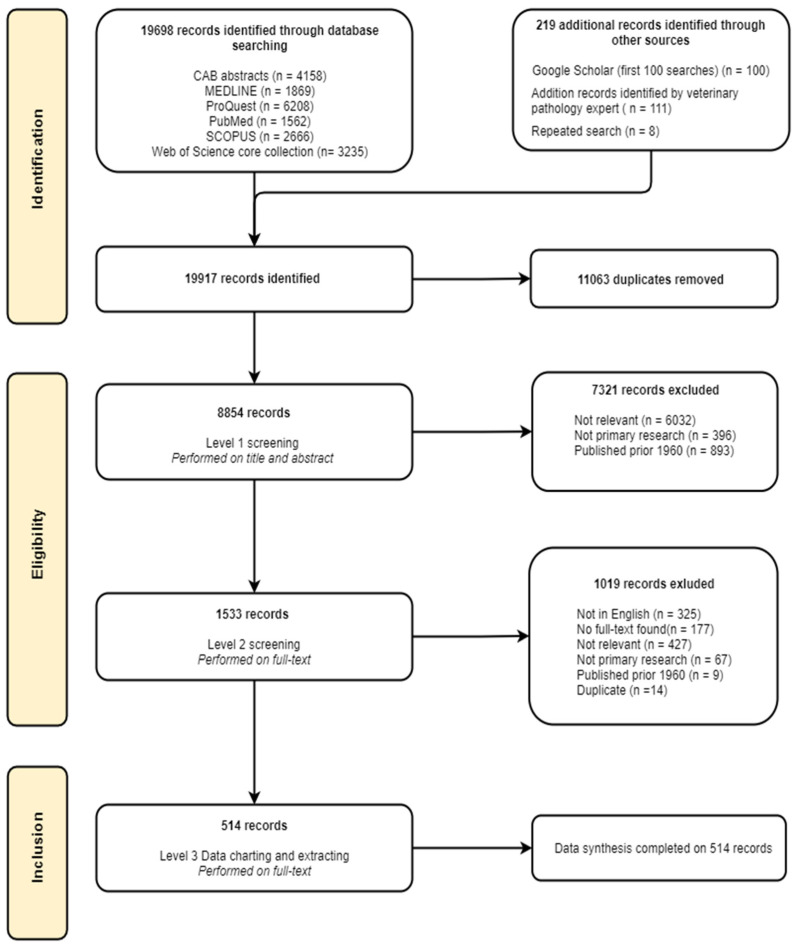
Flow chart of the numbers of studies identified, screened, assessed for eligibility, and included in a scoping review of causes and syndromes associated with mid to late-term pregnancy loss in horses.

**Figure 2 vetsci-09-00186-f002:**
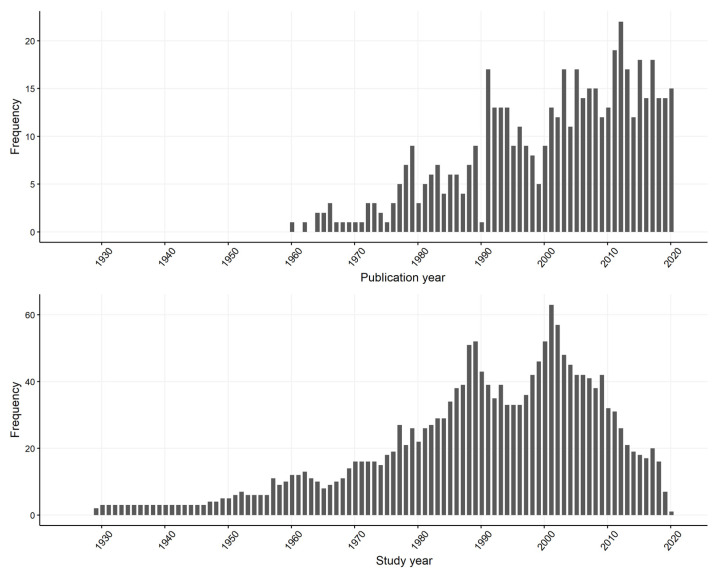
Distribution of the 514 studies captured from a scoping review of reported causes and syndromes associated with mid to late-term pregnancy loss in horses by (**top**) year of publication and (**bottom**) year of data collection from 1960–2020.

**Figure 3 vetsci-09-00186-f003:**
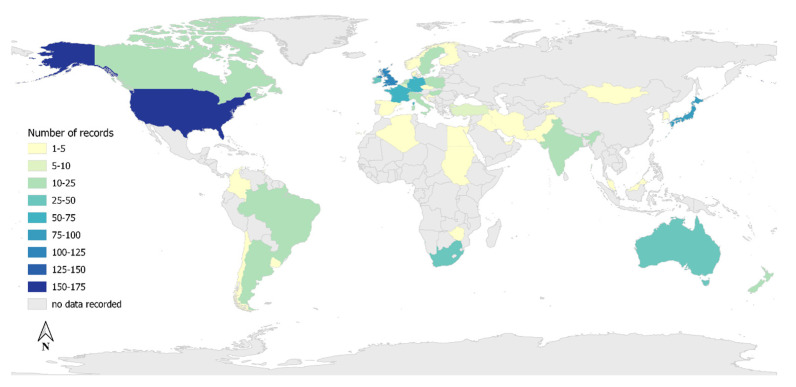
Location of data collected from the 446 records captured from a scoping review on the reported causes and syndromes associated with mid to late-term pregnancy loss in horses.

**Figure 4 vetsci-09-00186-f004:**
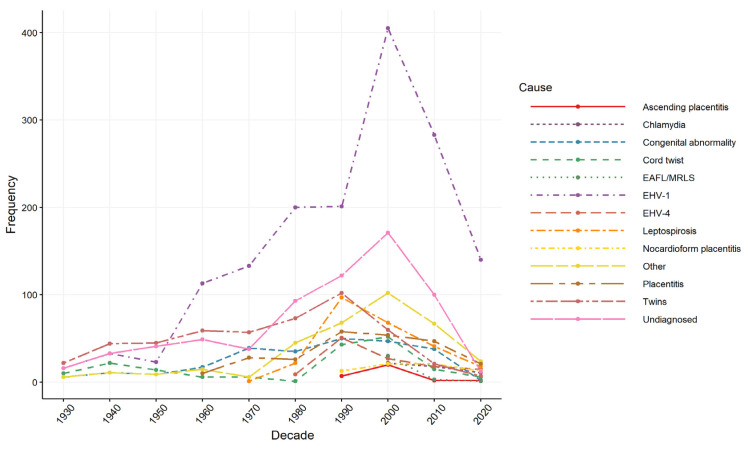
Causes of equine pregnancy loss by the decade of data collection from 514 studies captured from a scoping review of the reported causes and syndromes associated with mid to late-term pregnancy loss in horses.

**Figure 5 vetsci-09-00186-f005:**
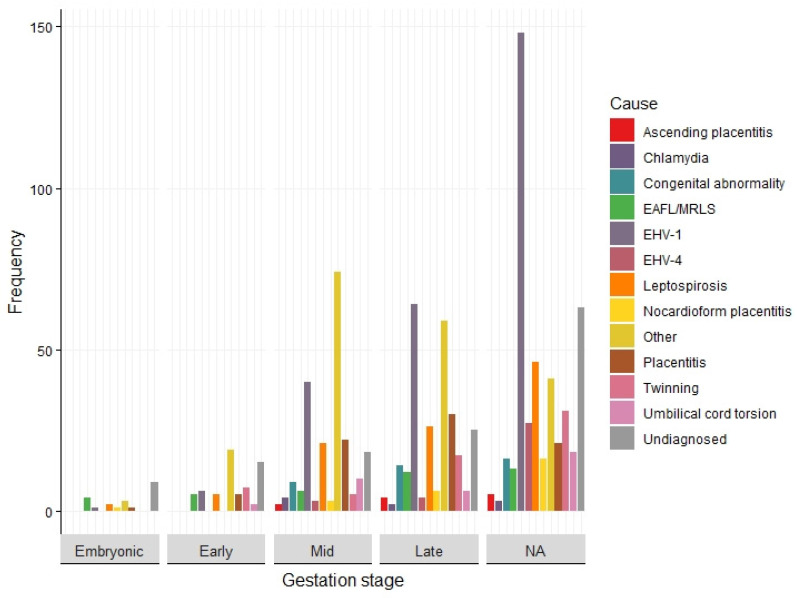
Causes and syndromes associated with equine pregnancy loss by stage of gestation collected from 514 studies captured in a scoping review of the reported causes and syndromes associated with mid to late-term pregnancy loss in horses globally. NA indicates records which did not provide data on the stage of gestation.

**Figure 6 vetsci-09-00186-f006:**
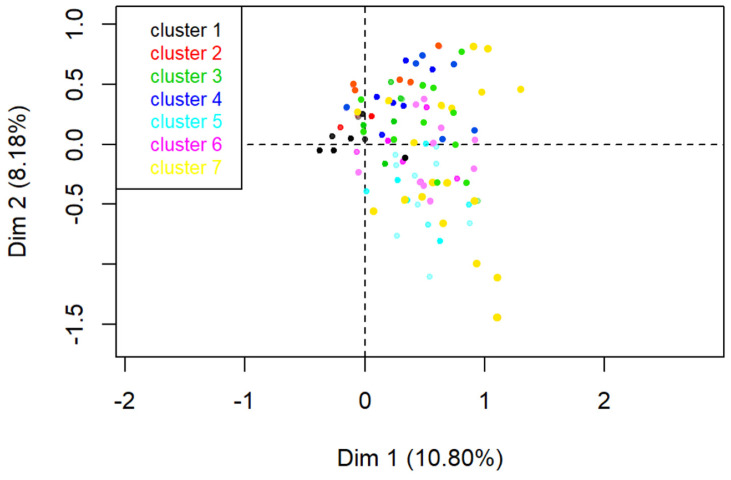
Ascending hierarchical classification of the individuals revealed 7 clusters from the MCA of data extracted from 514 studies captured in a scoping review of the reported causes and syndromes associated with mid to late-term pregnancy loss in horses globally.

**Table 1 vetsci-09-00186-t001:** The active and supplementary variables contributing to the formation of the clusters from the MCA of data extracted from 514 studies captured in a scoping review of the reported causes and syndromes associated with mid to late-term pregnancy loss in horses globally.

	Cluster 1	Cluster 2	Cluster 3	Cluster 4	Cluster 5	Cluster 6	Cluster 7
**Active variables**							
EAFLS/MRLS				yes			
EHV-1	yes	no	no	no	no	no	no
EHV-4							
Chlamydia							yes
Congenital abnormality							yes
Diagnosed cause of loss	yes	yes			no		
Leptospirosis	no	yes			no		
Placentitis	no			yes		yes	yes
Other bacteria	no	no		yes			
Other environmental causes			yes				
Other foetal causes				yes			
Other fungi							yes
Other maternal causes			yes				
Other placental causes			yes				
Other protozoa						yes	
Other virus		no	yes				
Twins		no			yes		
Umbilical Cord torsion						yes	
**Supplementary variables**							
Thoroughbreds		yes			yes		
Other Breed	yes		yes			yes	yes
Stage of gestation	no		mid/late		embryonic/early	early	mid
Global region	Europe	North America		North America	Asia	South America/No region	
Gross pathology on foetus				yes			yes
Gross pathology on foetal membranes	no		yes	yes		yes	yes
Histopathology on foetus		yes		yes	no	yes	
Histopathology on foetal membranes	no		yes	yes	no		yes

## Data Availability

The data presented in this study are available in article and Appendix A.

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
