# Peer review of "A Scoping Review of the Global Distribution of Causes and Syndromes Associated with Mid- to Late-Term Pregnancy Loss in Horses between 1960 and 2020"

_vetsci, 2022, doi:10.3390/vetsci9040186_

Round 1
Reviewer 1 Report
Comments
This manuscript is a scoping review of the global distribution of causes and syndromes associated with mid to late-term pregnancy loss in horses between 1960 and 2020.
This is more a statistical study than a clinical study. I am a clinician and I reviewed the clinical part and the scientific soundness of the manuscript. I recommend another reviewer for the statistical part (Preferred Reporting Items for Systematic reviews and Meta-Analyses extension for Scoping Reviews, PRISMA-ScR).
The study is not particularly original, but the objective is well achieved: identify the global spatiotemporal distribution of reported causes of mid to late-term equine pregnancy loss and identify the gaps in the current literature.
This study is well structured: materials and methods and results are clearly described, discussion considers some relevant arguments, conclusions are consistent with the proposed objectives.
Corrections
In the title and text of the scoping review you always say "causes and syndromes in pregnancy loss", however in the results and discussions you report "causes" but not "syndromes". Can you review the manuscript for this issue?
In the title and text of the scoping review you say “causes and syndromes associated with mid to late-term pregnancy loss...”…, however in results and in the identification of clusters 5 and 6 is considered embryonic and early foetal loss (lines 242, 244, 308, 310, 447, Fig 5, Table 1). You can check this part?
Line 82: peer-reviews records have a greater scientific soundness than without peer-reviewed records, however much bibliography from years gone by would be lost
Line 131: delete =(16)
Lines 181-182: in level 2 were included 1533 records. After exclusion were eligible 514 records. If now you repeat search for recent publications and include 8 records in level 3, records become 522. The 8 new records probably were included before (see figure 1 identification)
Line 196: add data collected “since 1930”
Lines 202-209: the result of records addition is 518, not 514
Lines 213-226: can you make a figure of this data as well?
Table 5 : Could you explain in the text the meaning of NA?
Line 306: Cluster 4: compared to the table 1 is missing “histopathology on the foetus”
Line 308: Cluster 5: “ with histopathology conducted on the foetus.“ What does it mean? yes or no? In table 1 is no
Line 315: Cluster 7 in the text: ” Gross and histopathology on the foetal membranes” ; Cluster 7 in table 1 “Yes for Gross pathology on foetus, Gross pathology on the foetal membranes and Histopatology on foetus”. Which one is right?
Line 372: I do not find correspondence with data in table 1 (early gestation)
Lines 410-426: can you reorganize the paragraph? The long discussion of zoonoses seems to refer only to Toxoplasma and to West Nile Virus
Lines 431-432: I do not find correspondence between lines 431-432 and table 1 and lines 308. which one is right? I think the one that considers late twin pregnancy loss
Lines 451-452: I do not find correspondence about umbilical cord torsion between lines 451-452 and the data in table 1 and lines 309-311
Lines 454-455: I do not find correspondence about umbilical cord torsion between lines 454-455 and the data in table 1 and lines 309-310
Author Response
Comment 1:
In the title and text of the scoping review you always say "causes and syndromes in pregnancy loss", however in the results and discussions you report "causes" but not "syndromes". Can you review the manuscript for this issue?
Response: We apologise for this omission and have added syndromes throughout the text where appropriate.
Comment 2:
In the title and text of the scoping review you say “causes and syndromes associated with mid to late-term pregnancy loss...”…, however in results and in the identification of clusters 5 and 6 is considered embryonic and early foetal loss (lines 242, 244, 308, 310, 447, Fig 5, Table 1). You can check this part?
Response: Thank you for asking for clarification on this issue. Some records providing information on pregnancy losses throughout gestation which could not be be simplified into only mid and late term losses and required reporting embryonic and early pregnancy loss.
Comment 3:
Line 82: peer-reviews records have a greater scientific soundness than without peer reviewed records, however much bibliography from years gone by would be lost
Response: We have updated the manuscript to explain why records were sourced from after 1960, this was due to changes in histopathology identification and changes the nomenclature. We agree that peer-reviewed records have greater scientific soundness, and for this reason, it was an eligibility criterion for this review. This change can be found at lines 84-86
Comment 4:
Line 131: delete =(16)
Response: We have corrected this error.
Comment 5:
Lines 181-182: in level 2 were included 1533 records. After exclusion were eligible 514records. If now you repeat search for recent publications and include 8 records in level 3, records become 522. The 8 new records probably were included before (see figure 1 identification)
Response: Thank you for pointing this out, this is correct. The repeat search for recent publications occurred before the final level of screening and these 8 publications were part of the 514 records. This change can be found at line 122.
Comment 6:
Line 196: add data collected “since 1930”
Response: We have updated the manuscript to include this. This change can be found at line 210-211
Comment 7:
Lines 202-209: the result of records addition is 518, not 514
Response: Thank you for pointing this out, we have corrected this. This change can be found at line 209-210
Comment 8:
Lines 213-226: can you make a figure of this data as well?2
Response: This information was captured in Figure 5, showing the frequency of causes. This can be found at lines 265-268
Comment 9:
Table 5 : Could you explain in the text the meaning of NA?
Response: Thank you for this comment, we have updated the manuscript to explain that NA indicates records which did not provide data on stage of gestation. This change can be found at line 267-268.
Comment 10:
Line 306: Cluster 4: compared to the table 1 is missing “histopathology on the foetus”
Response: Thank you for pointing this out, we have updated the manuscript. This change can be found at line 321.
Comment 11:
Line 308: Cluster 5: “ with histopathology conducted on the foetus.“ What does it mean? yes or no? In table 1 is no
Response: We agree that this sentence was confusing, we have updated the manuscript to be clearer for readers that no histopathology was conducted on the foetus. This change can be found at line 308
Comment 12:
Line 315: Cluster 7 in the text: ” Gross and histopathology on the foetal membranes” ; Cluster 7 in table 1 “Yes for Gross pathology on foetus, Gross pathology on the foetal membranes and Histopatology on foetus”. Which one is right?
Response: Thank you altering us to this issue, we have corrected the manuscript and table. This change can be found at line 330-331.
Comment 13:
Line 372: I do not find correspondence with data in table 1 (early gestation)
Response: This information can be found in the paragraph after Figure 3 (lines 229-242). We have added percentages to the numbers for the common causes.
Comment 14:
Lines 410-426: can you reorganize the paragraph? The long discussion of zoonoses seems to refer only to Toxoplasma and to West Nile Virus.
Response: Thank you for this comment, we have updated the manuscript to be clearer for readers that this section is about zoonotic diseases in general and is not exclusively describing only to Toxoplasma and to West Nile Virus. This change can be found at line 435.
Comment 15:
Lines 431-432: I do not find correspondence between lines 431-432 and table 1 and lines 308. which one is right? I think the one that considers late twin pregnancy loss
Response: We have updated the manuscript to include the gestation period of risk. The clusters were based on the most common factors to the least common. In this cluster undiagnosed and twinning were most commonly co-reported; however, these records also included information about embryonic and early gestation losses. This change can be found at lines 449-450.
Comment 16:
Lines 451-452: I do not find correspondence about umbilical cord torsion between lines 451-452 and the data in table 1 and lines 309-311
Response: Thank you for alerting us to this issue. The clusters were based on the most common factors to the least common. In this cluster, protozoa and twins were the most strongly co-reported. Although these records were also clustered with records on early gestation and South America, they may not necessarily have been to do with umbilical cord torsion.
Comment 17:
Lines 454-455: I do not find correspondence about umbilical cord torsion between lines 454-455 and the data in table 1 and lines 309-310
Response: Thank you for this comment, we have updated the manuscript to be clearer for readers, by separating the sentence. This change can be found at lines 472.

Reviewer 2 Report
Dear Dr. Macleay and co-workers,
Thank you for submitting this interesting piece of work for consideration for publication in Veterinary Sciences. A very interesting data set has resulted from your scoping review, which will be novel to many veterinary researchers, and this warrants publication in my opinion.
General comments:
I think it is likely that the timing of publication and the timing of the search strategy resulted in the exclusion of the following paper: Shilton, C.A., Kahler, A., Davis, B.W. et al. Whole genome analysis reveals aneuploidies in early pregnancy loss in the horse. Sci Rep 10, 13314 (2020). https://doi.org/10.1038/s41598-020-69967-z I can’t see a reason why it would not meet the search and inclusion criteria otherwise?
I have no specific comment for this reference to be considered other than to suggest that if the process actually missed it, it may be worth stating that the methodology is not absolute.
Specific comments:
Line 5: The co-author Angela Begg does not appear to have an affiliation attributed
Lines 99-101: A series of typos involving *
Line 130: Formatting; additional space between ‘…abstract, one’
Lines 129-131: ‘(n=16)’ is this un-nescessary or is the number supposed to be a sum of 42+1+16 = 59?
Lines 211-212: ‘Location of data collected from the 514 studies captured from a scoping review on the reported causes and syndromes associated with mid to late-term pregnancy loss in horses.’ This is a very small point however the way the data is presented pictorially against the countries should one state that this is the location data from 446 records rather than 514 studies?
Figure 4: Some of the line colours are too similar to easily differentiate even when viewing at higher magnification due to resolution of figure, consider colour and line form change.
Figure 5: I am not clear what the NA category is? Does it represent the 41% of records where gestation was not recorded? Please revise figure legend.
Line 283: ‘…were pregnancy loss due to other placental causes’ and…’
Author Response
Comments from Reviewer 1:
Comment 1:
I think it is likely that the timing of publication and the timing of the search strategy resulted in the exclusion of the following paper: Shilton, C.A., Kahler, A., Davis, B.W. et al. Whole genome analysis reveals aneuploidies in early pregnancy loss in the horse. Sci Rep 10, 13314 (2020). https://doi.org/10.1038/s41598-020-69967-z I can’t see a reason why it would not meet the search and inclusion criteria otherwise?
Response: Thank you for pointing this out. We agree with this comment, and it's probable that some references might be missed, despite our best efforts to capture all records. We have updated the manuscript to include this in the limitations. This change can be found at lines 476-478.
Comment 2:
The co-author Angela Begg does not appear to have an affiliation attributed
Response: We have updated the manuscript to include the current affiliations for Angela Begg.
Comment 3:
‘Location of data collected from the 514 studies captured from a scoping review on the reported causes and syndromes associated with mid to late-term pregnancy loss in horses.’ This is a very small point however the way the data is presented pictorially against the countries should one state that this is the location data from 446 records rather than 514 studies?
Response:
We agree with this and have removed the double up on the number of records. This change can be found at line 223.
Comment 4:
Figure 4: Some of the line colours are too similar to easily differentiate even when viewing at higher magnification due to resolution of figure, consider colour and line form change.
Response:
We have modified Figure 4 to increase the line thickness and have changed the colour and style of the graph so that it is easier to differentiate the causes. This change can be found at line 248
Comment 5:
Figure 5: I am not clear what the NA category is? Does it represent the 41% of records where gestation was not recorded? Please revise figure legend.
Response:
We agree with this and have updated the figure caption to make this clearer. This change can be found at lines 263-266
Reviewer 3 Report
Overall, this review is thorough and informative and provides interesting insight into equine abortion etiologies. There were a few minor points to address:
- Most equine literature considers a conceptus as transitioning from embryo to fetus at day 40. Was there a reason that you chose day 45 instead? This is a very minor curiosity- do you have a reference to support why you chose your specific cut off days for the four categories of pregnancy loss?
- Why was 1960 the cut off year? The discussion said that the reports of equine abortion increased at this time, though 893 records were excluded because they were published before 1960. Just explaining why this was chosen somewhere in the text would be helpful.
- For Figure 4, is there a way to make the lines thicker or change the shapes used on the lines? The colors are very challenging to distinguish.
- Another limitation that should be mentioned in the discussion is that the number of reported cases doesn't necessarily correlate directly with prevalence. This is sort of described when you discuss that Coxiella and Chlamydia are not well reported, but perhaps this should be mentioned more directly.
- Two minor typos:
- line 308: 'early gestation and HAD did not report EHV-1'
- line 415: Listeria monocytogenes is missing its second 'o'
Author Response
Comments from Reviewer 2:
Comment 1:
Most equine literature considers a conceptus as transitioning from embryo to fetus at day 40. Was there a reason that you chose day 45 instead? This is a very minor curiosity- do you have a reference to support why you chose your specific cut off days for the four categories of pregnancy loss?
Response:
Thank you for this comment. It is the most common practice within the horse breeding industry to perform ultrasound scanning between days 40-45, which is used to confirm whether the pregnancy has reached beyond the embryonic phase. We decided to include current industry practice into our definitions. We have updated the section on definitions to include a more detailed description. This change can be found at lines 91-101.
Comment 2:
Why was 1960 the cut off year? The discussion said that the reports of equine abortion increased at this time, though 893 records were excluded because they were published before 1960. Just explaining why this was chosen somewhere in the text would be helpful
Response:
We agree with this and have incorporated your suggestion. The reason for the choice was that a specialist in equine histopathology (AB) recommended that the record were sourced after 1960 due to changes in histopathology identification and changes the nomenclature. This change can be found at lines 83-84.
Comment 3:
For Figure 4, is there a way to make the lines thicker or change the shapes used on the lines? The colors are very challenging to distinguish.
Response:
We have modified Figure 4 to increase the line thickness and have changed the colour and style of the graph so that it is easier to differentiate the causes. This change can be found at line 248
Comment 4:
Another limitation that should be mentioned in the discussion is that the number of reported cases doesn't necessarily correlate directly with prevalence. This is sort of described when you discuss that Coxiella and Chlamydia are not well reported, but perhaps this should be mentioned more directly
Response:
We agree with this and have incorporated your suggestion. This change can be found at line 484-487.